# Correlation of host inflammatory cytokines and immune-related metabolites, but not viral NS1 protein, with disease severity of dengue virus infection

Hui Jen Soe[1], Rishya Manikam[2], Chandramathi Samudi Raju[1], Mohammad Asif Khan[3], Shamala Devi Sekaran[4]*

1 Department of Medical Microbiology, Faculty of Medicine, University of Malaya, Kuala Lumpur, Malaysia,
2 Trauma and Emergency (Academic), Faculty of Medicine, University of Malaya, Kuala Lumpur, Malaysia,
3 Centre for Bioinformatics, School of Data Sciences, Perdana University, Serdang, Selangor, Malaysia,
4 Faculty of Medicine and Biomedical Sciences, MAHSA University, Selangor, Malaysia

* shamaladevi@mahsa.edu.my

**Data Availability Statement:** All relevant data are within the manuscript and its Supporting Information files.

## Abstract

Severe dengue can be lethal caused by manifestations such as severe bleeding, fluid accumulation and organ impairment. This study aimed to investigate the role of dengue non-structural 1 (NS1) protein and host factors contributing to severe dengue. Electrical cell-substrate impedance sensing system was used to investigate the changes in barrier function of microvascular endothelial cells treated NS1 protein and serum samples from patients with different disease severity. Cytokines and metabolites profiles were assessed using a multiplex cytokine assay and liquid chromatography mass spectrometry respectively. The findings showed that NS1 was able to induce the loss of barrier function in microvascular endothelium in a dose dependent manner, however, the level of NS1 in serum samples did not correlate with the extent of vascular leakage induced. Further assessment of host factors revealed that cytokines such as CCL2, CCL5, CCL20 and CXCL1, as well as adhesion molecule ICAM-1, that are involved in leukocytes infiltration were expressed higher in dengue patients in comparison to healthy individuals. In addition, metabolomics study revealed the presence of deregulated metabolites involved in the phospholipid metabolism pathway in patients with severe manifestations. In conclusion, disease severity in dengue virus infection did not correlate directly with NS1 level, but instead with host factors that are involved in the regulation of junctional integrity and phospholipid metabolism. However, as the studied population was relatively small in this study, these exploratory findings should be confirmed by expanding the sample size using an independent cohort to further establish the significance of this study.

## Introduction

Dengue, a self-limiting illness accompanied by high fever, headache, muscle and joint pains, vomiting or rash, is a mosquito borne viral disease transmitted by female mosquitoes of the

**Funding:** SDS received grant from University of Malaya Research Grant Grant No: RP042C-15HTM https://umresearch.um.edu.my/funding-opportunities-information SHJ received grant from University Malaya Postgraduate Research Grant Grant No: PG255-2015B. https://umresearch.um.edu.my/funding-opportunities-information The funders had no role in study design, data collection and analysis, decision to publish, or preparation of the manuscript.

**Competing interests:** The authors have declared that no competing interests exist.

species *Aedes aegypti* or *Aedes albopictus* [1]. The causative agent, dengue virus (DENV), is a positive-sense single-stranded RNA virus belonging to the family *Flaviviridae* which comprises of 4 distinct but closely related serotypes named DENV-1, DENV-2, DENV-3 and DENV-4 [2]. Although dengue can be asymptomatic or presented mostly as an acute mild febrile illness, 0.05% of the cases progressing to severe dengue had potentially fatal complications such as plasma leakage, severe bleeding or organ impairment [3]. Vascular leakage, the hallmark of severe dengue usually involves the loss of function or increased permeability of the endothelium that lines the pleural and peritoneal cavities [4]. The pathogenesis of severe dengue is theorized to be due to the intricate interactions between viral factors, host genetics and host immune activation [5]. Dengue viral proteins such as the structural and non-structural proteins have been demonstrated to differentially modulate host immune responses, including antibody neutralization activity [6], complement alternative pathway [7], T cell activation [8, 9] and the production of pro-inflammatory cytokines [10–12], which play a major role in the enhancement of the disease.

The RNA genome of DENV is translated into a long polypeptide that is further cleaved and processed by viral and host proteases into three structural proteins (envelope, capsid protein and precursor membrane) and seven non-structural proteins (NS1, NS2A, NS2B, NS3, NS4A, NS4B and NS5) [13]. Being one of the most conserved and enigmatic protein among the non-structural proteins, NS1 is critically required for RNA replication and the production of infectious viral particles [14]. It is secreted into the blood circulation of the host during dengue virus infection and has been used as a viral marker for early diagnosis of dengue for decades [15]. However, NS1-induced vascular leakage was discussed only in recent years and the contribution of NS1 to vascular leakage as well as the mechanism of pathogenesis remains controversial. Several studies reported that the level of NS1 was higher in patients with severe dengue (SD) [16, 17] and it appears to contribute to disease severity by inducing IL-10 production by monocytes [18], activating immune cells via toll-like receptor 4 [19] or disrupting the endothelial glycocalyx components leading to increased vascular permeability [20]. On the other hand, studies also showed that low level of NS1 can also be associated with more severe manifestations [21] and the NS1 positivity did not correlate with severe pathologies [22]. These contradictory findings obscure the role of NS1 in inducing vascular leakage observed in patients with SD, leaving an unaddressed research gap for further studies.

The immunopathogenesis of severe dengue is mediated mainly by humoral immune response supported by the evidence of exaggerated cytokine production from antibody-dependent enhancement and complement activation by antigen-antibody complexes [23]. Although individuals with secondary infections are believed to have a higher risk of developing severe dengue due to the presence of cross-reactive non-neutralising antibodies from primary infection, severe manifestations have also been reported in patients with primary infections [24] with increased level of inflammatory cytokines [25, 26]. The infecting serotypes also contribute to the risk of severe dengue, for instance, infection with DENV-3 resulted in a greater percentage of severe cases for primary infection in Southeast Asia (SEA) region, while for secondary infection, DENV-2, DENV-3 and DENV-4 from SEA region, as well as DENV-2 and DENV-3 from the non-SEA regions displayed greater percentages of severe cases [24]. Despite the immune status of the host, abundant evidences suggested that high levels of inflammatory cytokines contribute to the development of severe manifestations in DENV infection [27, 28]. The production of unfavourable cytokines from massive immune activation of monocytes, macrophages and T cells play an important role in causing endothelial dysfunction and increased vascular permeability. However, contradictory findings have also been reported on the correlation between cytokine levels and disease severity in dengue patients. Several cytokines such as IFN-γ, TNF-α, IL-4, IL-10, IP-10 and VCAM-1 have been shown to be elevated

in patients with SD [29–31] while in other studies, the same cytokines such as IFN-γ and TNF-α showed no difference in levels between patients with mild dengue fever and severe dengue [27, 32]. Despite the ambiguous role of inflammatory cytokines in determining disease severity, the potential use of certain cytokines as biomarkers to predict the progression of severe dengue has been highlighted [33, 34]. This intention emphasized the importance of research on dengue immunopathogenesis for successful identification of biomarkers that are expressed differentially in severe dengue compared to other categories of dengue.

Since endothelial cells have been proposed to be prominently involved in the immuno-pathogenesis of plasma leakage and haemorrhagic manifestations in severe dengue [35], it is important to be able to investigate the real time changes of the barrier function of the cells *in vitro* under the simulation of DENV infection. An electrical-substrate impedance sensing (ECIS) technique has been used for this purpose. ECIS is a real time approach to monitor barrier function or permeability of cell monolayers electrically using different frequencies of alternative current (AC) for constant measurement. High frequency capacitance signifies the establishment of a confluent cell layer and a low frequency resistance represents the formation of para-cellular passage, where the combination of both parameters into impedance, using the mathematical model in ECIS software, provides an insight towards the barrier function of the cells. An increase in impedance would indicate junctional tightening and a reduction would indicate a loss of barrier function, which in our case would indicate a vascular leakage as endothelial cells were used [36].

Hence, this study aims to investigate the role of the viral factor, specifically the NS1 protein and the host factors, in term of the expression of inflammatory cytokines and immune-related metabolites in contributing to the degree of severity of dengue virus infection. We investigated the effect of NS1 at varying levels to the barrier function of microvascular endothelium using ECIS, as well as the effect of NS1 in the presence of other host factors in patient sera from three categories of disease severity including dengue without warning signs, dengue with warning signs and severe dengue, in causing vascular leakage in microvascular endothelium. After determining the clinical relevance of viral factor (NS1) in inducing vascular leakage, we assessed the cytokine expression profile and the metabolomics profile of these patients in comparison to the expression profile of healthy individuals to identify host factors that may underlie dengue disease processes.

## Materials and methods

### Sample collection and selection

A total of 40 archived serum samples previously collected under ethical statement of IRB reference number 926.4 from UMMC Medical Research Ethics Committee (MREC) with written informed consent were used in this study. Among the 40 samples, 30 samples were from anonymized patients diagnosed with dengue virus infection, while 10 samples were collected from volunteered healthy individuals. The clinical data were recorded during blood collection and the disease severity was categorized into dengue without warning signs (DWOWS), dengue with warning signs (DWWS) and severe dengue (SD) according to the guidelines on case classification provided by WHO in 2009 (S1 Table) [3]. The serum samples were subjected to DENV reverse transcriptase polymerase chain reaction (RT-PCR) with iTaq™ Universal SYBR® Green One-step Kit (Bio-rad Laboratories, Hercules, CA, USA), Panbio Dengue Early ELISA (Alere, Waltham, MA, USA), SD BIOLINE Dengue IgM capture ELISA (Alere) and hemagglutination inhibition (HI) test to confirm dengue virus infection. For ECIS measurement, five samples were selected randomly for each dengue category from a panel of NS1 and IgM positive samples to assess the effect of NS1 levels by setting the presence of IgM antibodies

as a constant variable. Five samples that were dengue positive from laboratory tests were selected randomly and a total of 10 samples including those used for ECIS measurement for each dengue category were included for cytokine and metabolomics profiling. Ten samples from healthy individuals were selected and confirmed with negative results from laboratory tests.

## Cell culture

Three microvascular endothelial cell lines including primary human dermal microvascular endothelial cells (HDMEC) (Sciencell, Carlsbad, CA, USA), primary human pulmonary microvascular endothelial cells (HPMEC) (Sciencell) and primary human retinal microvascular endothelial cells (HRtMEC) (CSC, Tysons, VA, USA) were cultured as *in vitro* models for dengue virus infection. The cells were maintained at 37˚C with 5% $CO_2$ supplemented with endothelial cell growth medium (ECM) (Sciencell) for HDMEC and HPMEC, and CSC complete medium (CSC) for HRtMEC.

## Electrical cell-substrate impedance sensing (ECIS)

The three MEC lines were grown to a confluent monolayer in an eight-well electrode array (8W10E+) (Applied Biophysics, Troy, NY, USA) in triplicates prior to the treatment with NS1 antigen or serum samples for ECIS measurement. The NS1 antigen (Bio-rad) used is a recombinant dengue virus serotype 1 NS1 protein of sequence strain Nauru/Western Pacific/1974 expressed in 293 human cells. It is produced with a C-terminal 6x His-tag and at a purity of >95% by SDS PAGE analysis diluted in buffer solution Dulbecco's phosphate buffered saline, which was further diluted during the experiment using the respective media of each cell line used. High (1ng/mL), medium (10pg/mL) and low (0.1pg/mL) levels of recombinant NS1 antigen were selected based on the values obtained in NS1 ELISA test that corresponds to the level of NS1 detected in the selected serum samples. The cells were treated with NS1 antigens at the three selected concentrations, five serum samples that had varying levels of NS1 antigen from each category (DWOWS, DWWS, SD) and five healthy individuals (HC). The real time changes in impedance (resistance and capacitance) induced in the MECs after treatment was detected using multiple alternative current (AC) frequencies in an ECIS Zθ 16-well array station (Applied Biophysics) for 72 hours. The raw data was first normalized to cell-free wells to eliminate background signals, followed by normalization to 0 hours post infection (hpi) in the untreated cells. The standard error of mean between the triplicates was determined and presented at regular intervals.

## Cytokine profiling

The level of 14 commonly studied cytokines, chemokines, adhesion molecules and growth factors (CCL2, CCL5, CCL20, CD25, CXCL1, CXCL6, CXCL9, CXCL10, CXCL11, IL-18, TNF-a, VEGF-A, HMGB-1 and ICAM-1) in the 40 selected serum samples were assessed using the Human Magnetic Luminex multiplex screening assay based on flow cytometric analysis of magnetic antibody-coated microbeads targeting the analytes of interest (R&D Systems, Minneapolis, MN, USA). These 14 cytokines have been shown to have a role during dengue virus infection in terms of endothelial immune activation [34, 37, 38] and among them, CCL2, CCL5, CCL20, CXCL1, CXCL10, CXCL11, TNF-α and ICAM-1 have been reported to be produced directly by DENV-infected microvascular endothelial cells [39]. The precision and reproducibility of the assays were assured by calculating the coefficient of variance of the replicates. The significant differences of the cytokine expression profile between the categories were determined using student's T test.

## Untargeted metabolomics

The metabolic profiling analysis of the 40 selected serum samples was conducted using quadrupole time of flight liquid chromatography mass spectrometry (QTOF-LCMS) technique. Protein precipitation and metabolite extraction from serum samples was performed by adding 3 volumes of cold (-20˚C) mixture of methanol and ethanol (volume ratio of 1:1) to 1 volume of serum. Samples were vortex-mixed and stored at -20˚C for 5 minutes, followed by centrifugation at 16,000 × g at 4˚C for 10 minutes. The pellet was removed and the supernatant was filtered through a 0.22μm nylon filter and subjected to LCMS procedure.

10μL of extracted serum sample was applied to a reversed-phase column (Agilent 959764–902, Eclipse Plus C18, 2.1x100mm, 1.8um, 600Bar) (Agilent Technologies, Santa Clara, CA, USA) in QTOF-LCMS system (Agilent 6550 iFunnel) operated at a flow rate of 0.6mL/min in positive electrospray ionization (ESI) mode. The solvents used were solvent A, water with 0.1% formic acid and solvent B, acetonitrile with 0.1% formic acid. The gradient started from 25% B to 95% B in the first 35 minutes, returned to 25% B within the next 1 minute, and maintained at 25% B for the remaining 9 minutes. Samples were analysed in one randomized run with a capillary voltage of 3000V and nebulizer gas flow rate of 10.5L/min. During the analysis, two reference masses: 121.0509 m/z ($C_5H_4N_4$) and 922.0098 m/z ($C_{18}H_{18}O_6N_3P_3F_{24}$) were continuously measured to allow constant mass correction. Data were required in positive ion mode with a full scan from 50 to 1000 m/z at a rate of 1.02 scans per second.

The raw data files obtained from QTOF-LCMS were extracted using an untargeted batch-processing feature extraction software, MassHunter Profinder (Agilent) to generate processed .cef file to be input into a chemometrics software, Mass Profiler Professional (Agilent) for data analysis described below [40]. Unsupervised principle component analysis (PCA) was performed for quality control of the samples based on the variability in the data and their possible correlation. The included data set was then reordered and grouped into HC, DWOWS, DWWS and SD. Data was prepared by filtering the frequency and abundance, aligning the parameters based on tolerances established by retention time and mass, normalizing to reduce the variability caused by sample preparation and instrument response followed by base-lining the statistical abundance across all the samples for further data analysis. Fold change of the metabolic expression in dengue groups (DWOWS, DWWS and SD) against HC was calculated and statistical analysis was performed using multivariate Welch's one-way ANOVA with Benjamini Hochberg false discovery rate for multiple group comparison. Cut-off value of fold change > 2.0 with p-value < 0.05 was used for selection. Selected metabolites were highlighted by calculating the prevalence of expression among the categories along with the abundance of ion detection representing the level of expression in each individual. Visualizations such as Venn diagrams, volcano plots, hierarchical complete linkage clustered heat maps, histograms and boxplots have been constructed to present the findings of the experiment. Pathway enrichment analysis was also completed based on BioCyc Database [41] and Human Metabolome Database [42] to identify metabolic pathways involved in the progression of dengue virus infection.

## Results

### Samples characteristics

The clinical data and diagnostic results of twenty serum samples selected for ECIS measurement is shown in Table 1. All the samples from HC category were DENV PCR negative, NS1 negative, IgM negative but contained low concentration of neutralizing antibodies (<10, 10, 20). For dengue positive samples, all were DENV PCR negative with varying levels of NS1 ranging

**Table 1. The results for DENV PCR, NS1 ELISA, IgM ELISA and HI for five serum samples from each category accompanied by the day of illness.** Panbio Units were calculated based on manufacturer's protocol with a cut off value of >11 for NS1 positive samples and <9 for NS1 negative samples.

| SAMPLES | CATEGORY | DENV PCR | NS1 ELISA | PANBIO UNIT | IGM ELISA | HI | DAY OF ILLNESS |
|---|---|---|---|---|---|---|---|
| H1 | HC | - | - | 1.85 | - | <10 | - |
| H2 | HC | - | - | 2.56 | - | 20 | - |
| H3 | HC | - | - | 1.90 | - | 20 | - |
| H4 | HC | - | - | 2.00 | - | 10 | - |
| H5 | HC | - | - | 1.82 | - | <10 | - |
| WO1 | DWOWS | - | + | 75.78 | + | <10 | 3 |
| WO2 | DWOWS | - | + | 69.80 | + | 20 | 4 |
| WO3 | DWOWS | - | + | 58.28 | + | 2560 | 6 |
| WO4 | DWOWS | - | + | 56.40 | + | 640 | 6 |
| WO5 | DWOWS | - | + | 27.90 | + | 20 | 3 |
| W1 | DWWS | - | + | 75.66 | + | 2560 | 6 |
| W2 | DWWS | - | + | 69.91 | + | <10 | 6 |
| W3 | DWWS | - | + | 54.50 | + | 10 | 5 |
| W4 | DWWS | - | + | 39.77 | + | 10 | 5 |
| W5 | DWWS | - | + | 38.22 | + | <10 | 5 |
| S1 | SD | - | + | 69.02 | + | 2560 | 3 |
| S2 | SD | - | + | 67.35 | + | 20 | 4 |
| S3 | SD | - | + | 65.92 | + | 10 | 7 |
| S4 | SD | - | + | 58.34 | + | 80 | 4 |
| S5 | SD | - | + | 39.6 | + | 20 | 6 |

from 27.90 to 75.78 Panbio Unit. Eleven samples have a low HI titre at <10, 10, 20 and 80, while four samples contained high level of neutralizing antibodies with HI titre of 640 and 2560. The day of illness of dengue patients ranged from 3–7 days at the time of collection. The clinical diagnosis of 40 serum samples selected for cytokine and metabolomics profiling is summarized in Table 2. Additional five samples from each category have been added to the panel on top of the previous selected samples for ECIS study. The number of DENV PCR positive samples were one and two in categories of DWOWS and DWWS respectively. Eight of DWOWS, six of DWWS and six of SD samples were NS1 positive, while DWOWS, DWWS and SD consist of nine, eight and nine IgM positive samples, respectively. The day of illness at the time of collection ranged from 3–6 days in DWOWS, 4–6 days in DWWS and 4–7 days in SD.

**Table 2. The number of samples tested positive for DENV PCR, NS1 ELISA and IgM ELISA among the 10 serum samples selected for each dengue category accompanied by the day of illness.** These ten samples consisted of five previously selected samples from ECIS measurement and five serum samples randomly selected from a pool of dengue positive samples, as well as for the healthy individuals.

| CATEGORIES | TOTAL SAMPLE NUMBER | DENV PCR POSITIVE | NS1 ELISA POSITIVE | IGM ELISA POSITIVE | DAY OF ILLNESS |
|---|---|---|---|---|---|
| HC | 10 | 0 | 0 | 0 | - |
| DWOWS | 10 | 1 | 8 | 9 | 3–6 |
| DWWS | 10 | 2 | 6 | 8 | 4–6 |
| SD | 10 | 0 | 6 | 9 | 4–7 |

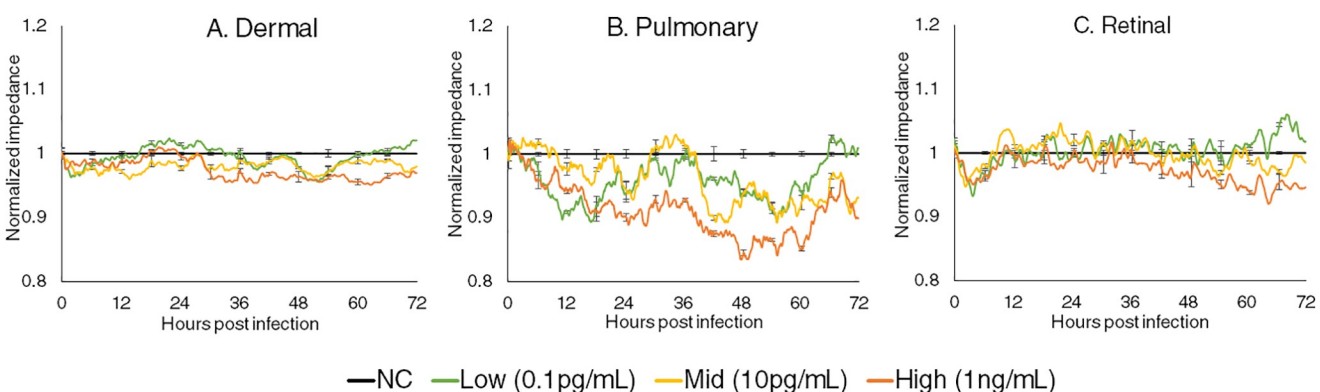

**Fig 1.** Normalized real-time impedance changes in (A) Dermal MEC, (B) Pulmonary MEC and (C) Retinal MEC treated with low (0.1pg/mL), medium (10pg/mL) and high (1ng/mL) levels of NS1 antigens. The measured impedances of the treated MECs were normalized to the un-treated cells and the standard error of mean between biological triplicates were represented as error bars.

### NS1 modulates MECs barrier function in a dose dependent manner

The NS1 concentration of the fifteen serum samples selected for ECIS measurement were categorized into high (Panbio Unit 60–80), medium (Panbio Unit 40–60) and low (Panbio Unit 20–40) levels. The NS1 ELISA results for 1ng/mL, 10pg/mL and 0.1pg/mL of NS1 antigen are 68.7, 51.2 and 32.1 Panbio Unit respectively, representing high, medium and low level of NS1 corresponding to the level of NS1 detected in the serum samples.

The changes in the impedance of the MECs treated with three levels of NS1 were measured using ECIS system for 72 hpi and the results are presented in Fig 1. Dermal and retinal MECs responded in a similar manner, starting with a slight impedance reduction upon the introduction of NS1, which then increased at 12 hpi, followed by a fluctuating but progressive reduction from 24 hpi onwards. A small difference in the modulation was observed only at 60 hpi onwards between the MECs treated with three different levels of NS1, where higher level of NS1 reduced the impedance of MECs at a higher magnitude. The changes of impedance in pulmonary MECs were dose dependent. In pulmonary MECs treated with high levels of NS1, the impedance reduced drastically upon introduction of NS1 for up to 60 hours before increasing. The MECs treated with low and medium level of NS1 also showed a reduction in impedance at 0 hpi, followed by an increase at 12 hpi and 24 hpi respectively. The impedance eventually reduced at 36 hpi and recovered to control levels at 72 hpi.

### The loss of microvascular barrier function in severe dengue did not correlate with NS1 level

The five serum samples randomly selected from each category for the ECIS measurement consisted of samples with high (Panbio Unit 60–80), medium (Panbio Unit 40–60) and low levels (Panbio Unit 20–40) of NS1. Dermal, pulmonary and retinal MECs were incubated with all serum samples for 72 hours and the changes in impedance triggered by NS1 protein contained in the serum samples were assessed using ECIS system to investigate the modulation to MECs barrier function.

Fig 2 shows the changes in impedance of dermal MECs representing the modulation of barrier function after the incubation with five serum samples from each of the HC, DWOWS, DWWS and SD categories. In all categories, impedance reduced immediately upon the introduction of sera, followed by an increase at 1 hpi which then peaked at 7 hpi. For HC, the

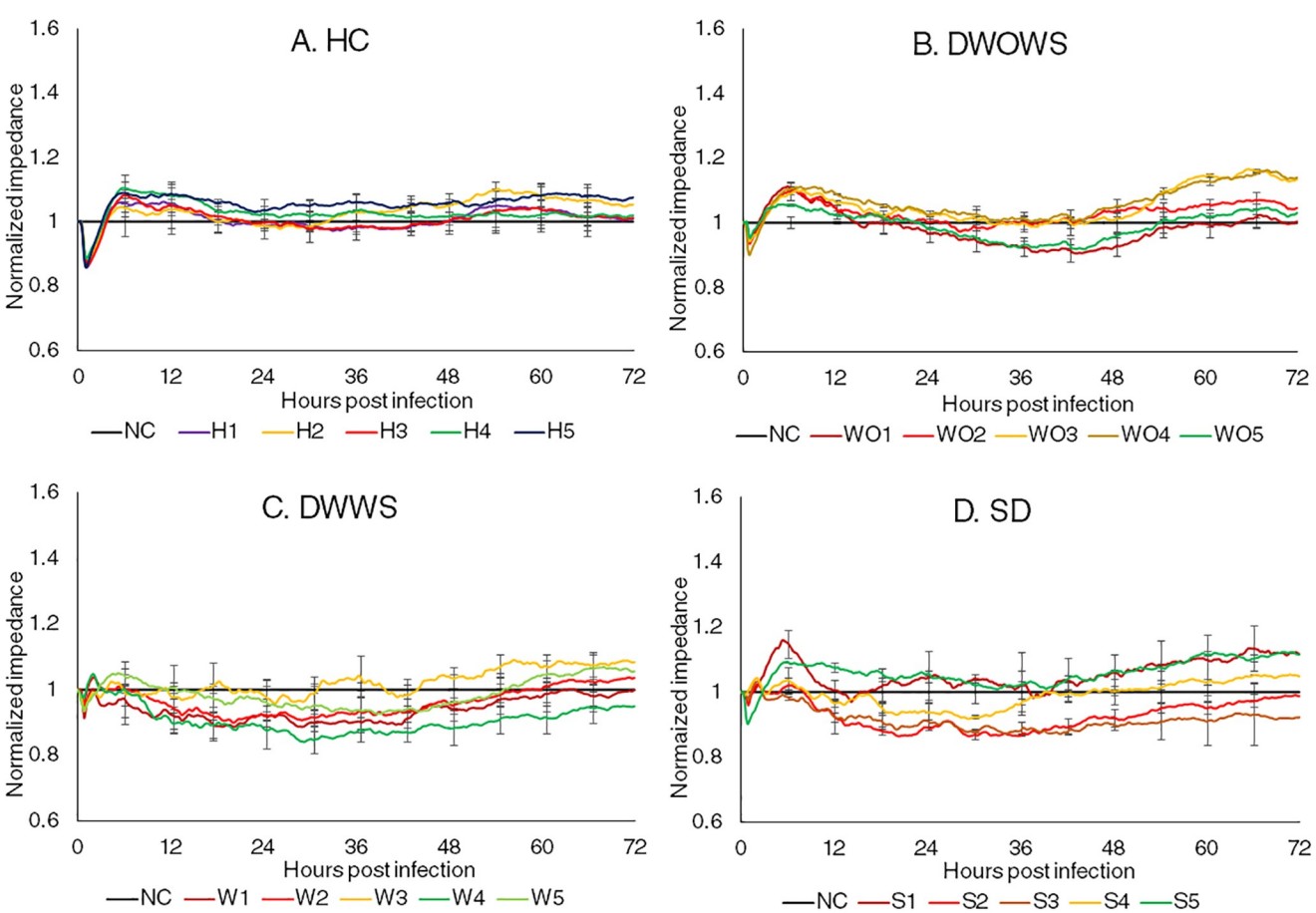

**Fig 2.** Normalized real time impedance changes in <u>dermal MECs</u> incubated with serum samples from the category of (A) Healthy individuals (HC), (B) dengue without warning signs (DWOWS), (C) dengue with warning signs (DWWS) and (D) severe dengue (SD). Five serum samples from each dengue category consisted of samples with high NS1 level (red lines), medium NS1 level (yellow lines) and low NS1 level (green lines). The error bars represent the standard error of mean between three biological triplicates for each sample.

impedance then maintained at a similar level as compared to the non-treated control cells, while for the other three dengue categories, most of the samples had an impedance reduction below control level indicating vascular leakage and fluctuated across the 72 hours. However, no distinct changing pattern was observed in between MECs incubated with sera at different levels of NS1 as well as sera from different categories.

Fig 3 shows the changes in impedance measured from pulmonary MECs incubated with 20 serum samples, five from each of the categories of HC, DWOWS, DWWS and SD. Upon introduction of sera, the MECs responded in a similar manner in all samples, starting with a slight increase in impedance and peaked at 1–2 hpi, followed by a reduction in impedance. However, the pattern varied between categories. In HC, the impedance fluctuated but maintained at a similar or above the level of the untreated cells. In DWOWS, all five samples increased the impedance of the MECs across 72 hours regardless of the level of NS1 in the samples. In DWWS, the changing pattern varied between samples, however the level of leakage did not correlate with NS1 level. MECs incubated with sample W1 with high NS1 concentration maintained its impedance at a similar level with the untreated cells, while another sample with high NS1 level (sample W2) induced a reduction in impedance below control level up to 72 hours,

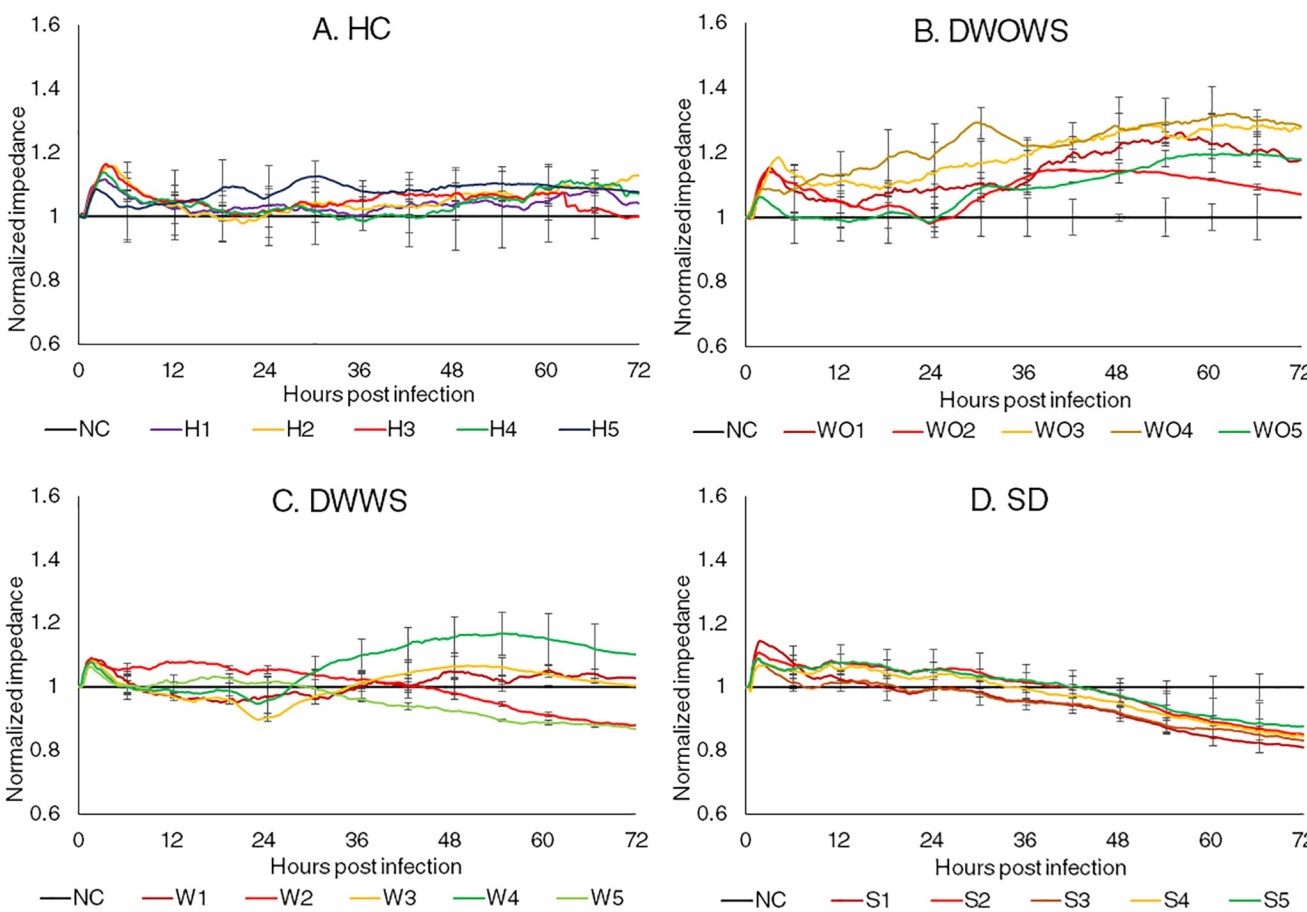

**Fig 3.** Normalized real time impedance changes in pulmonary MECs incubated with serum samples from the category of (A) Healthy individuals (HC), (B) dengue without warning signs (DWOWS), (C) dengue with warning signs (DWWS) and (D) severe dengue (SD). Five serum samples from each dengue category consisted of samples with high NS1 level (red lines), medium NS1 level (yellow lines) and low NS1 level (green lines). The error bars represented the standard error of mean between three biological triplicates for each sample.

showing a similar modulation with MECs incubated with a sample of low NS1 level (sample W5). The two remaining samples which consisted of one sample with medium NS1 level (sample W3) and one sample with low NS1 level (sample W4), showed a similar modulation to the MECs where the impedance reduced up to 24 hours, followed by an increase of impedance over control levels across the remaining 36 hours. In SD, the MECs incubated with all sera showed a similar changing pattern in impedance, where the impedance reduced gradually after 1 hpi and eventually dropped below control levels from 36 hpi onwards.

The changes in impedance of retinal MECs incubated with five serum samples from each category are presented in Fig 4. Similar to other two MECs, cells treated with sera from HC did not differ much with untreated cells. Samples from DWOWS induced an increase in the impedance upon the introduction of the sera at a higher magnitudes as compared to DWWS and SD, and the impedance reduced gradually but was maintained above the control levels across 72 hours. Samples with lower NS1 concentration did not modulate the MECs as much in comparison to samples with higher NS1 concentration, where the impedance fluctuated the least and maintained similarly to the untreated cells. However, the impact of NS1 in causing leakage is not conclusive as the samples from SD with low NS1 concentration was able to cause

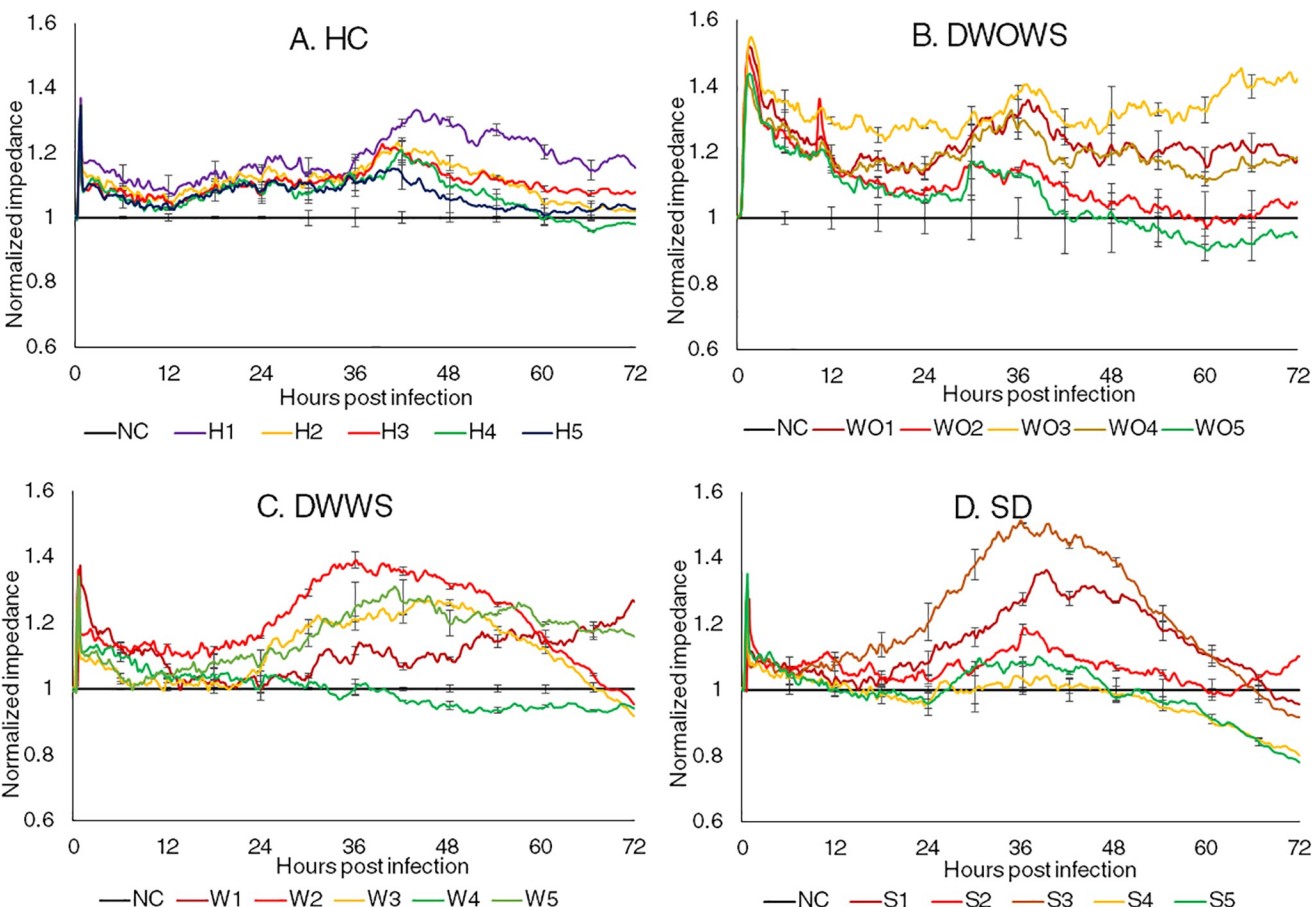

**Fig 4.** Normalized real time impedance changes in retinal MECs incubated with serum samples from the category of (A) Healthy individuals (HC), (B) dengue without warning signs (DWOWS), (C) dengue with warning signs (DWWS) and (D) severe dengue (SD). Five serum samples from each dengue category consisted of samples with high NS1 level (red lines), medium NS1 level (yellow lines) and low NS1 level (green lines). The error bars represent the standard error of mean between three biological triplicates for each sample.

leakage at 60 hpi, but the samples with high NS1 level in the same category induced leakage later at 68 hpi.

## Cytokines highly expressed in dengue patients did not correlate with disease severity

Cytokine profiling was performed on 40 serum samples from four categories, HC, DWOWS, DWWS and SD (Fig 5). Most of the studied cytokines including CCL2, CCL5, CCL20, CD25, CXCL1, CXCL5, CXCL9, CXCL10, CXCL11, IL-18, TNF-α and VEGF-A, were highly expressed in dengue patients as compared to healthy individuals. The mean of concentrations of CCL2, CCL20, CXCL1, CXCL6, CXCL9 and TNF-α was higher in patients with SD, while CCL5, CXCL10, CXCL11 and IL-18 were expressed at lower levels in patients with SD, however, the differences were not significant.

## Severity-specific expression of HMGB-1 and ICAM-1 in dengue patients

Cytokine/chemokine HMGB-1 and adhesion molecule ICAM-1 were expressed significantly higher in patients with SD and the comparison is shown in Fig 6. The mean concentration of

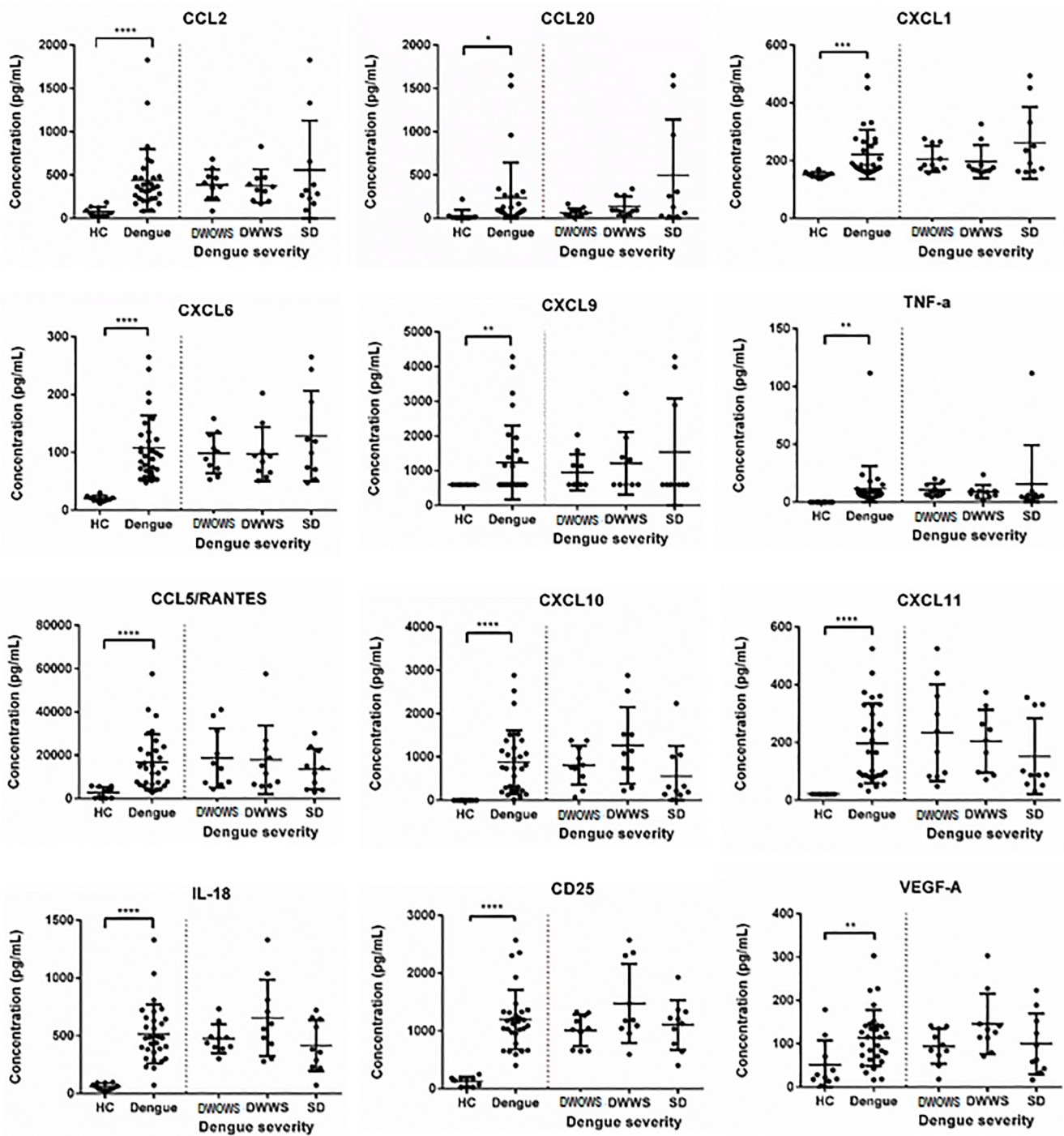

**Fig 5. Comparison of serum levels of cytokines, chemokines, adhesion molecules and growth factors in HC and dengue patients, as well as patients in the categories of DWOWS, DWWS and SD.** Student's T test was used to compare between HC and dengue cases, then across all three patient groups for significant difference represented by a p value less than 0.05, where *<0.05, **<0.01, ***<0.001 and ****<0.0001.

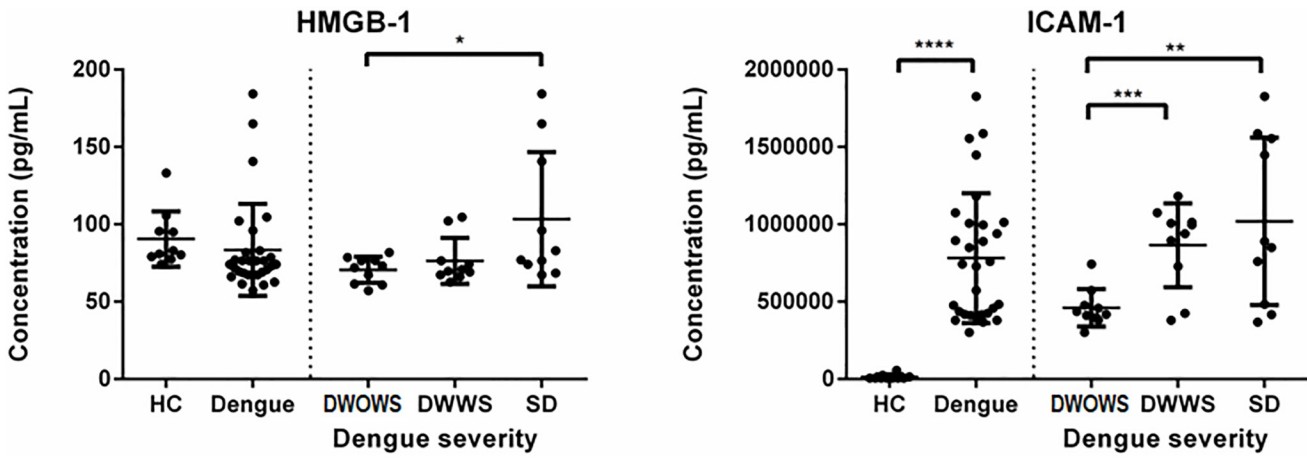

**Fig 6. Comparison of serum levels of HMGB-1 and ICAM-1 in HC and dengue patients that were further categorized into DWOWS, DWWS and SD according to disease severity.** The significant differences in the expression level between the categories were determined using Student's T test and a p value of less than 0.05 representing significantly lower or higher, indicated by *<0.05, **<0.01, ***<0.001 and ****<0.0001.

HMGB-1 was highest in patients with SD, however, the difference in expression between HC and dengue cases were not significant, making the role of this cytokine in dengue pathogenesis inconclusive. On the other hand, ICAM-1 was expressed significantly higher in dengue cases than in healthy individuals, and the mean concentration was significantly higher in SD patients as compared to DWOWS and DWWS.

## Metabolic expression in dengue patients

QTOF-LCMS was performed to compare the pattern of differential metabolic expression among DWOWS, DWWS and SD categories. Two dimensional PCA analysis revealed no outliers and the samples in the same category were tightly clustered. A total of 35,042 compounds were detected from 40 samples, where 1377 compounds were known metabolites identified based on their molecular weight and retention time crosschecked with Agilent database. Fold change (FC) analysis and one-way ANOVA was performed and a total of 500 metabolites met the standard of $FC > 2.0$ and $p < 0.05$ subjected to hierarchical clustering by complete linkage. The expression and regulation of metabolites present in the serum from each category is summarized in supplementary S1 Table. The heat map of 40 samples (Fig 7iA) showed that HC (blue) and DWOWS (green) were clustered together, separated from the cluster of DWWS (yellow) and SD (red). The clustered heat map based on four categories (Fig 7iB) showed that the same cluster of metabolites were up-regulated while different cluster of metabolites were down-regulated in the three dengue categories (DWOWS, DWWS and SD).

Filtering by volcano plot (Fig 7ii) revealed the differential metabolic expression of DWOWS, DWWS and SD as compared to HC in term of FC and significant difference. A total of 215, 239 and 220 metabolites were up-regulated in DWOWS, DWWS and SD, respectively, while a smaller number of metabolites were down-regulated in all categories (17 in DWOWS, 4 in DWWS and 15 in SD).

The up-regulated and down-regulated metabolites were further analysed into Venn diagrams (Fig 7iii), where metabolites that were expressed only in each category (DWOWS, DWWS and SD) were identified. As many as 49, 47 and 47 metabolites were up-regulated only in DWOWS, DWWS and SD respectively, while DWOWS, DWWS and SD specifically down-

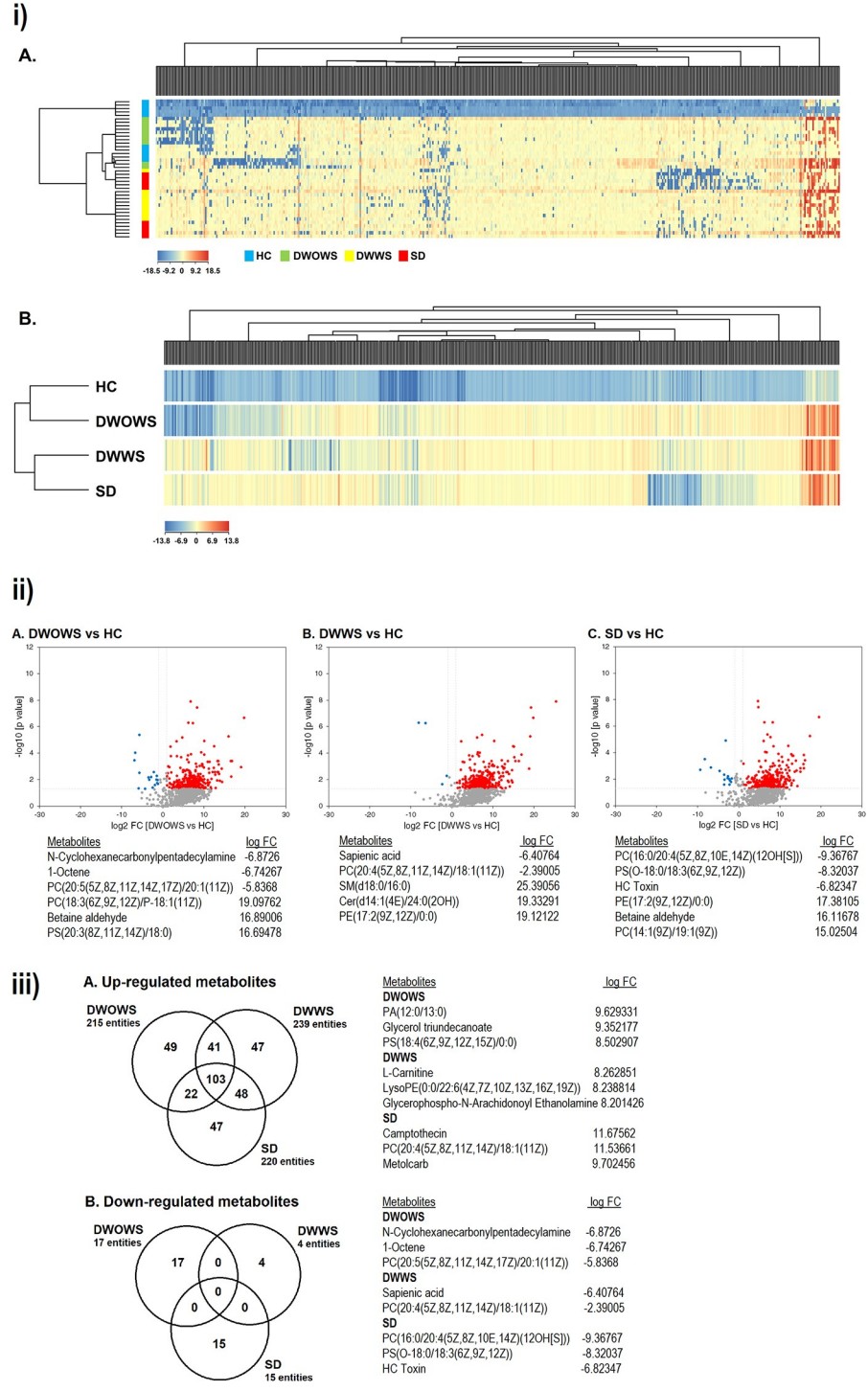

**Fig 7.** i) Hierarchical complete linkage clustered heat map of 500 metabolites met the requirement of FC > 2.0 and p < 0.05. Clustering based on individual samples (A) and categories (B) revealed the similarity as well as the differential metabolic expression in HC/DWOWS and DWWS/SD. ii) Volcano plot comparing p value against fold change of (A) DWOWS, (B) DWWS and (C) SD as compared to HC, and the top regulated metabolites in each category. A higher number of metabolites were up-regulated in dengue patients from all three categories compared to down-regulated metabolites. iii) Venn diagrams of (A) up-regulated and (B) down-regulated metabolites in DWOWS, DWWS and SD as compared to HC, and the top regulated metabolites in each category. DWOWS, DWWS and SD shared high commonality in the up-regulated metabolites, however, no metabolite was commonly down-regulated in all the categories.

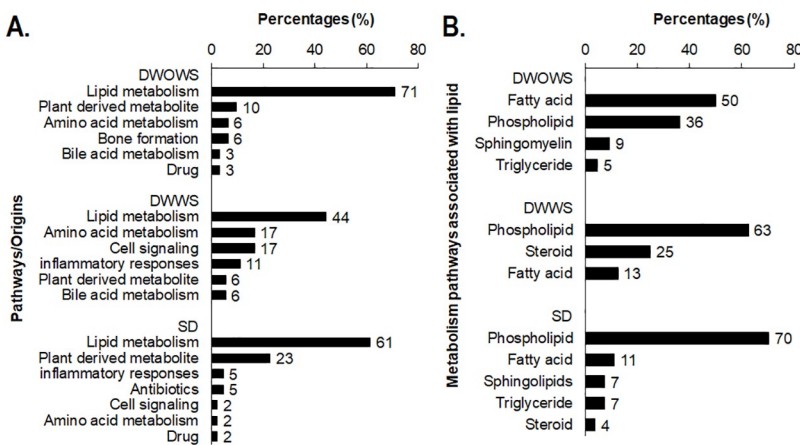

**Fig 8. Pathway analysis of metabolites specifically regulated in patients from categories of DWOWS, DWWS and SD.** A) The number of metabolites involved in the pathways were calculated as percentage (%) of the total metabolites regulated in each category. B) Metabolites involved in lipid metabolism pathway were further categorized into lipid associated metabolism pathways.

regulated 17, 4 and 15 metabolites respectively, and no metabolite was commonly down-regulated in all the three categories.

Pathway analysis was performed on metabolites that were expressed specifically in each category of DWOWS, DWWS and SD and the most regulated pathways were arranged according to the percentages of metabolites involved in each pathway (Fig 8). The findings revealed that majority of the regulated metabolites are involved in lipid metabolism pathway. Of the total metabolites, 71% in DWOWS, 44% in DWWS and 61% in SD were involved in lipid metabolism pathway. In DWOWS, most of the metabolites were involved in bone formation and amino acid metabolism, and included metabolites that originate from plant and drug compounds. In DWWS, 17% of the metabolites were involved in cell signalling pathway for apoptosis, membrane permeability and ROS production. Another 17% of the metabolites were involved in amino acid metabolism including glycine, serine, betaine and methionine metabolism. In SD, 23% of the metabolites were plant derived, exogenous to the host. These metabolites include triterpenoids, isoquinolines, triacylglycerols, flavonoids, phenylpropenes and chalcones originated from plant such as bush banana, compositae flower, mountain toatoa, *Tiliacora racemosa*, *Camptotheca acuminata* and citrus, where some have been categorized as medicinal plant to have antibiotic and antioxidant properties. Further analysis of the metabolites in lipid metabolism pathway revealed that most of the metabolites (50%) in DWOWS involved fatty acid metabolism, while 63% and 70% of the metabolites in DWWS and SD respectively involved phospholipid metabolism.

The metabolites, namely L-isoleucine, L-phenylalanine, 2-amino-3-methyl-1-butanol, C16 sphinganine, indoleacrylic acid and neoabietic acid, were detected in most of the healthy individuals (70–100%), but had a lower prevalence in dengue patients (Fig 9). Among the six metabolites, L-isoleucine, L-phenylalanine, 2-amino-3-methyl-1-butanol and C16 sphinganine had the lowest prevalence in patients with SD (20–50%), and the level of expression of C16 sphinganine was significantly lower in SD when compared with DWWS with a p value of <0.05. Although 2-amino-3-methyl-1-butanol was not detected in all dengue patients, the concentration of this metabolite was significantly higher in those dengue patients with a p value of <0.01. On the other hand, the prevalence of neobietic acid was lower in dengue patients, however SD had the highest prevalence among the three dengue categories.

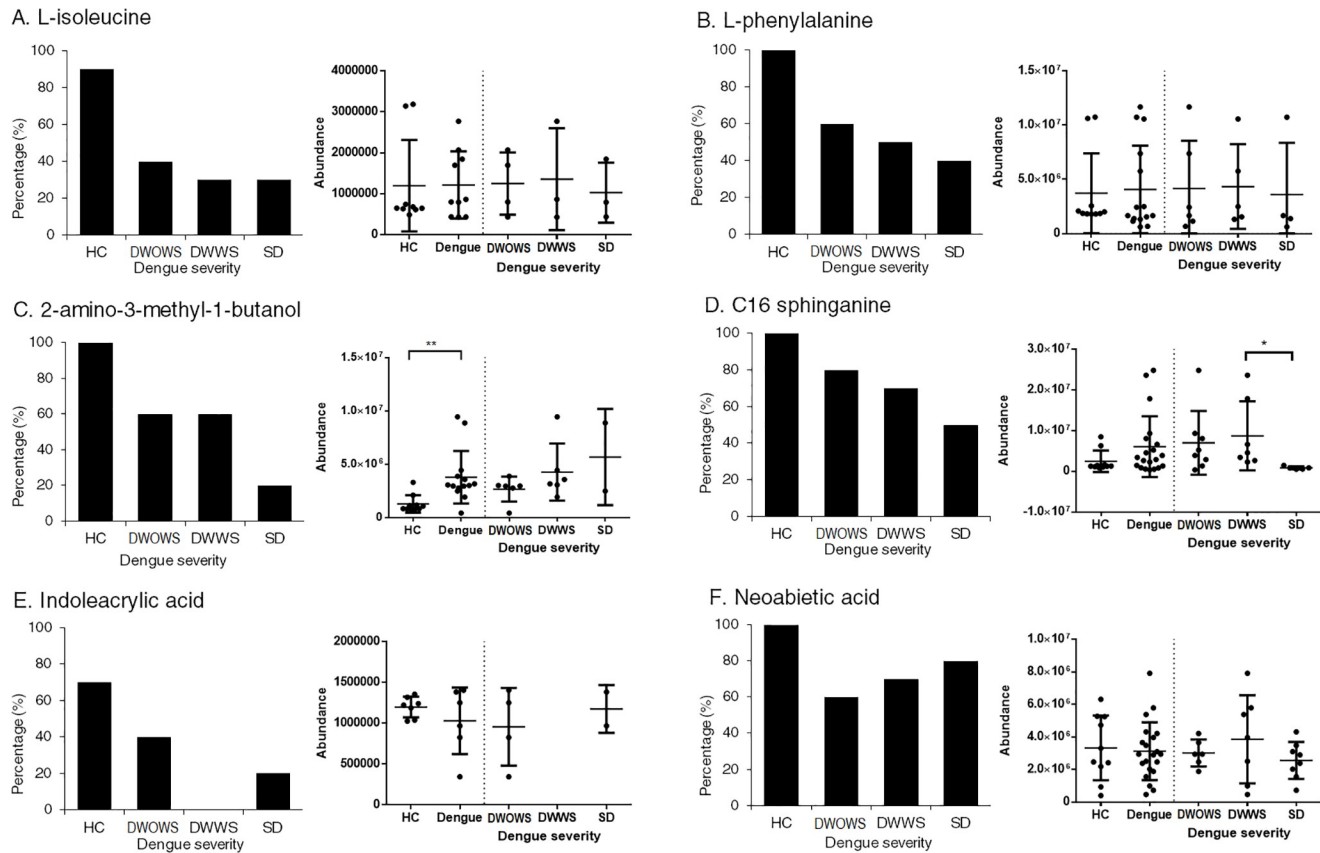

**Fig 9.** The percentage of patients expressing A) L-isoleucine, B) L-phenylalanine, C) 2-amino-3-methyl-1-butanol, D) C16 sphinganine, E) indoleacrylic acid and F) neoabietic acid and the abundance of expression of these metabolites for each individual in categories of HC, DWOWS, DWWS and SD. Student's T test was used to compare HC and dengue cases, as well as among the three dengue groups for significant difference in abundance, represented by a p value less than 0.05, where $^{*}<0.05$ and $^{**}<0.01$. Individuals with zero abundance are not shown on the dot plot.

Two metabolites, Etn-1-P-cer and Palmitic amide (Fig 10) were noted to be present at higher levels in dengue patients as compared to healthy individuals. Etn-1-P-cer was detected in 20% of the healthy individuals, 40% of both DWOWS and DWWS and in 70% of the patients with SD, while 50% of the healthy individuals, 70% of the patients with DWOWS and

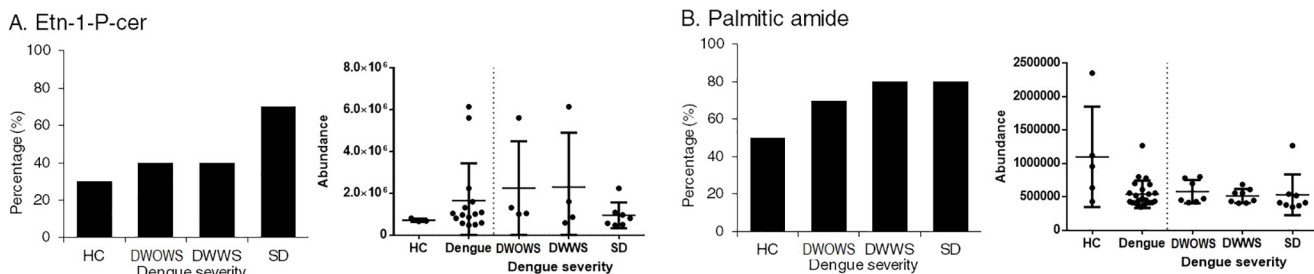

**Fig 10.** The percentages of patients expressing A) Etn-1-P-cer and B) palmitic amide, and the abundance of expression of these metabolites for each individual in the categories of HC, DWOWS, DWWS and SD. The significant difference in expression was determined using Student's T test to compare between HC and dengue cases as well as among the three dengue groups, however the differences were not significant with p values were more than 0.05. Individuals with zero abundance are not shown on the dot plot.

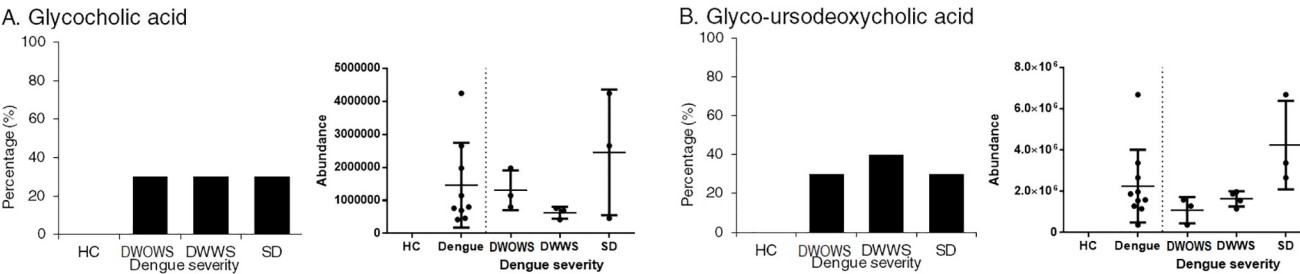

**Fig 11.** The percentages of patients expressing A) glycocholic acid and B) glycol-ursodeoxycholic acid and the abundance of expression of these metabolites for each individual in the categories of HC, DWOWS, DWWS and SD. Student's T test was used to determine the significant difference in expression among DWOWS, DWWS and SD, however the differences were not significant as the p values were more than 0.05. Individuals with zero abundance are not shown on the dot plot.

80% of both the patients with DWWS and SD had palmitic amide detected in their serum samples. However, the abundance of these metabolites detected was not significantly different among the three dengue categories.

Glycocholic acid and glyco-ursodeoxycholic acid was detected only in dengue patients but the prevalence was low ranging from 30–40% in each category (Fig 11). The mean concentration of these metabolites was higher in patients with SD, however the difference of expression was not significant as compared to DWOWS and DWWS.

## Discussion

Our study showed that NS1 alone can induce loss of barrier function in microvascular endothelium in a dose dependent manner. Based on ECIS measurement, the magnitude of vascular leakage observed was positively correlated with the concentration of NS1 protein, especially pronounced in pulmonary MECs. However, serum NS1 level in dengue patients did not directly correlate to the extent of vascular leakage observed in MECs, indicating that there might be other factors overshadowing the direct effect of circulating NS1 to the endothelium, but still leading to the vascular leakage observed in patients with SD. The association of NS1 antigen with disease severity in dengue virus infection have been commonly studied, however the results are contradictory and inconclusive. Several studies reported that the presence of circulating NS1 antigen was associated with a higher risk of developing severe manifestations and the serum NS1 levels were higher in patients with SD than those with mild dengue fever [16, 17, 43]. On the other hand, there is a paper showing that low NS1 antigen level instead was associated with more severe form of dengue [21]. Furthermore, studies also reported that there is no significant difference in NS1 concentration among patient groups of different disease severity [22], which is in agreement with our study, highlighting the complicated pathogenesis of dengue virus infection.

The loss of barrier function observed in the NS1-treated MECs has also been reported in other studies where NS1 was shown to degrade endothelial glycocalyx components [20, 44], increase the secretion of angiopoietin-2 and disrupt the function of junctional protein VE-cadherin [45] in endothelial cells, leading to hyper-permeability and endothelial dysfunction. However, the clinical effect of serum NS1 on endothelial barrier function has not been well studied and our results showed that the effect of NS1 in inducing vascular leakage was not prominent in clinical cases as compared to its direct stimulation on MECs, proposing a larger involvement of other factors such as cytokines or metabolites produced during dengue virus infection in the host.

To address the presence of factors other than NS1 protein in determining disease severity in clinical cases, 40 serum samples were subjected to cytokines and metabolomics profiling. Among all the studied cytokines, HMGB-1 and ICAM-1 were found to be correlated with disease severity, where the expression was significantly higher in patients with SD. HMGB-1 (high mobility group box 1) is a DNA binding protein that regulates innate immunity and partially the adaptive immunity [46]. This protein can be released into the extracellular environment passively upon apoptosis or necrosis of cells [47] or actively by activated macrophages and monocytes [48], which then functions as a pro-inflammatory cytokine in the inflammatory response positive feedback loop to stimulate endothelial cells for cytokine and chemokine production [49]. HMGB-1 had also been shown to induce the production of ICAM-1 in endothelial cells [50], which was also expressed significantly higher in patients with SD in our study. ICAM-1, an intercellular adhesion molecule, plays an important role in leukocytes transmigration at the sites of activated endothelial cells and the formation of immunological synapse during cellular immune responses [51, 52]. Previous studies have reported the roles of HMGB-1 and ICAM-1 in dengue virus infection where both proteins were released by dengue infected endothelium or immune cells [39, 53] and increased permeability of endothelial cells leading to vascular leakage [54, 55]. They were detected at a higher concentration in dengue patients [56, 57], while HMGB-1 was also detected in the peripheral organs of dengue fatal cases [38], which further justified the role of HMGB-1 and ICAM-1 in the progression of severe dengue.

Metabolomics study using QTOF-LCMS revealed that metabolites specifically regulated in DWOWS, DWWS and SD mostly interact with lipid metabolism pathway. DENV has been shown to mediate lipid synthesis and metabolism in their replication cycles [58] by taking advantage of the production of double membrane vesicles during autophagy for efficient replication [59]. In patients with DWOWS, most of the metabolites are involved in fatty acid metabolism for energy generation and to create triglycerides, phospholipids and other important membrane constituents [60], probably to maintain cellular processes to repair damages upon dengue virus infection [61]. On the other hand, patients with DWWS and SD expressed large number of metabolites that are involved in phospholipid metabolism pathway, which regulates the formation and function of the membrane bilayer [62]. Phospholipids are a class of lipids consist of a phosphate group that can form lipid bilayers and function as the major components of cell membrane [63]. Components from different class of phospholipids such as phosphatidylcholine, phosphatidylglycerol and phosphatidylserine as well as phosphatidic acid, which are precursors for other more complex phospholipids were expressed differentially in dengue patients, especially those with DWWS and SD. Deregulated phospholipid metabolism is likely to be due to the changes in exogenous intake of fatty acid from patient's diet during infection or altered activities of lipid-metabolizing enzymes induced by DENV [64]. Perturbations of the phospholipid metabolism might contribute to the destabilization of membrane permeability during dengue virus infection [65]. Furthermore, patients with SD specifically expressed metabolites originated from sphingolipid metabolism pathway, where sphingolipids are found mainly in the membranes of brain and nervous cells. Altered sphingolipid metabolism might lead to rearrangement of membrane components which has been associated with various neurological diseases [66] that might possibly link to the rare neurological complications, such as brachial neuropathy or encephalopathy observed in patients with SD [67].

Our metabolomics findings have also identified amino acids such as L-isoleucine, L-phenylalanine and 2-amino-methyl-1-butanol, which are essential in the biosynthesis of proteins were lacking in the sera of dengue patients, with the lowest prevalence in patients with SD. Changes in the plasma free amino acid pattern, indicating infection-related alterations of

amino acid metabolism, has been reported in dengue virus infection [68]. The disruption of protein biosynthesis pathway might be due to DENV-induced changes in the host cell transcriptional and translational machinery [69] involved along the DENV replication cycle [70]. Besides, cell membrane-associated metabolites such as C16 sphinganine and indoleacrylic acid were also found to have a lower prevalence in dengue patients. Interaction of DENV with cellular membranes for viral replication during infection has been widely reported [71]. Since membrane-associated metabolites modulate membrane integrity by altering the biophysical properties of cell membrane [72, 73], the differential expression of these metabolites in dengue patients might contribute to the loss of endothelial barrier function leading to vascular leakage observed in dengue patients.

Contrarily, some metabolites were detected at a higher prevalence in dengue patients, where healthy individuals expressed such metabolites at a lower occurrence or none at all. Etn-1-P-cer and palmitic amide, the membrane fatty acids in the family of sphingolipids, were detected higher in dengue patients, especially in patients with SD. Studies have shown that sphingolipids and other bioactive signalling molecules were up-regulated in DENV infected mosquitoes cells [65] and enhanced trans-endothelial cell permeability in DENV infected microvascular cells [74]. Sphingolipids consist of highly complex and distinct interconverting bioactive lipids that function as signalling molecules for cell migration, adhesion and inflammatory responses, which also form the chemically resistant and mechanically stable outer leaflet of the membrane bilayer [75, 76]. The abundance of bioactive lipids from sphingolipids in the patient's sera indicate a disruption of sphingolipids biosynthesis and metabolism, possibly contributing to metabolic disorders, immune dysfunction, skin integrity and in our case, perhaps vascular diseases [77]. Two metabolites involved in bile acid synthesis were also detected only in dengue patients, albeit at a low prevalence. The metabolism of glycocholic acid, a conjugated primary bile acid synthesized by the liver [78] and glycol-ursodeoxycholic acid, a secondary bile acid enzymatically modified by microbiota [79], was differentially expressed in dengue patients. Impaired bile acid homeostasis has been shown to contribute to the pathogenesis of liver diseases [80], including acute liver failure that has been viewed as one of the indication of severe dengue [81]. Furthermore, these two metabolites have a role in fat emulsification and cholesterol regulation; the interference in their metabolism might modulates cholesterol biosynthesis and change the lipid profile of dengue patients during infection [82, 83]. Overall, our findings are in agreement with a serum metabolomics investigation of humanized mouse model where the major perturbed pathways included sphingolipid, amino acid and bile acid metabolism during dengue virus infection [84].

The interplay between host immune responses and metabolic processes, especially the interactions between inflammation and lipid metabolism has been reported to exacerbate the development of several diseases including atherosclerosis [85], rheumatoid arthritis [86], as well as infections by pathogens such as *chlamydia trachomatis* [87] and hepatitis C virus [88]. Modified lipids, fatty acids and lipoproteins have shown to interfere with inflammatory response and directly affect immune cell activation [89, 90], while altered inflammatory signalling directly impacts lipid metabolism in metabolic and immunity diseases [91]. Although the interconnection between inflammatory cytokines and lipid-associated metabolites during DENV infection has not been well reported, the cytokines highly expressed in dengue patients from this study such as CCL2, CCL5, CCL20, CXCL1, CXCL5 and IL-18 have been shown to associate with the disruption of lipid metabolism in other diseases, however, most of the studies were done with a mouse model. The blockage of CCL2 receptor, CCR2 has been shown to induce phenotypic changes of adipocytes leading to improved lipid metabolism in the kidney of mice with diabetic nephropathy; the high level of CCL2 observed in DENV infection might interfere lipid metabolism through activation of CCR and its downstream pathway [92].

Furthermore, high level of IL-18 increased AMPK signalling leading to increased fat oxidation and might correlates with reduced concentration of circulating triglycerides [93, 94]. CXCL1, which is functional equivalent with IL-18, was also shown to enhance fatty acid oxidation at high concentration [95]. Although some of the cytokines have not been reported on its effect on lipid metabolism, they are known to associate with lipid-associated diseases. For instance, the expression of CCL5 and CCL20 has been associated with fatty liver disease [96, 97], while CXCL5 is an adipose tissue derived factor associated with obesity [98]. The interference of these cytokines with lipid metabolism in other diseases provides insights on the plausible alteration of lipid metabolism caused by cytokine storm during DENV infection, and the possibility of this leading to the disruption of cell membrane permeability merits further investigation.

One of the principle caveats for this study is the fact that ECIS measurement does not provide direct information at the molecular level as the readings are generated based on the interaction between electric current flows and cell morphology changes. However, this study focused only on the cellular level to compare the direct or indirect stimulation of NS1 in MECs, hence, it is thus informative at the beginning of our experimental series to obtain input for the construction of a feasible hypothesis. Another limitation of the study was the small sample size of only 40 sera due to the difficulties in obtaining more samples from patients with SD. Although the study has less statistical power, we were able to identify a few cytokines and metabolites that would provide insight on the role of host factors in the pathogenesis of dengue virus infection and possibly determine or predict disease severity that should be assessed using a larger sample size in the future.

## Conclusions

Our study revealed that dengue NS1 protein was able to induce the loss of barrier function of the microvascular endothelium in a dose dependent manner. However, the level of NS1 did not correlate with the extent of vascular leakage observed in the microvascular endothelium treated with serum samples from patients with dengue virus infection. This finding suggested the presence of other host factors that might overshadow the direct effect of NS1 in inducing vascular leakage during dengue virus infection. Most of the cytokines that were highly expressed in dengue patients including CCL2, CCL5, CCL20 and CXCL1 are involved in leukocyte infiltration where the rearrangement of junctional complex proteins such as ICAM-1, which was detected significantly higher in patients with SD, may lead to the disruption of inter-endothelial junctions. Altered lipid metabolism was detected in dengue patients, where severe manifestations were observed in patients with DWWS and SD. This might be associated with phospholipid metabolism, which may affect the membrane permeability of cells including the microvascular endothelium. Our study further highlighted the complexity of dengue as no single factor alone, either viral or host, can be responsible for the progression of severe dengue, nor can be used as prognostic marker for disease severity. In this circumstance, the identification of these altered metabolites could facilitate dengue diagnosis or be used as a potential target for new therapeutic options.

## Supporting information

**S1 Table. Metabolites present in the serum and its regulation comparing DWOWS, DWWS and SD with HC.**
(DOCX)

**S1 Fig.**
(TIF)

## Acknowledgments

We acknowledge Ms Heng Kai Yen and Ms See Hui Shien from Biomed Global in helping with the procedures of Human Magnetic Luminex multiplex screening assay. We also acknowledge Mr Lim Teck Maan from Agilent Crosslab Malaysia in providing assistance on using MassHunter Profinder and Mass Profiler Professional for the data analysis of metabolomics study.

## Author Contributions

**Conceptualization:** Chandramathi Samudi Raju, Shamala Devi Sekaran.

**Data curation:** Hui Jen Soe, Rishya Manikam, Shamala Devi Sekaran.

**Formal analysis:** Hui Jen Soe, Chandramathi Samudi Raju, Mohammad Asif Khan.

**Funding acquisition:** Shamala Devi Sekaran.

**Investigation:** Hui Jen Soe, Rishya Manikam, Shamala Devi Sekaran.

**Project administration:** Chandramathi Samudi Raju, Shamala Devi Sekaran.

**Resources:** Rishya Manikam.

**Software:** Mohammad Asif Khan.

**Supervision:** Chandramathi Samudi Raju, Mohammad Asif Khan, Shamala Devi Sekaran.

**Validation:** Hui Jen Soe, Mohammad Asif Khan, Shamala Devi Sekaran.

**Writing – original draft:** Hui Jen Soe, Shamala Devi Sekaran.

**Writing – review & editing:** Rishya Manikam, Chandramathi Samudi Raju, Shamala Devi Sekaran.

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
