## [Decision Letter · Decision Letter 0]

18 Dec 2019

PONE-D-19-24309

Correlation of viral NS1 protein, host inflammatory cytokines and immune-related metabolites with disease severity of dengue virus infection

PLOS ONE

Dear Professor Sekaran,

Thank you for submitting your manuscript to PLOS ONE. After careful consideration, we feel that it has merit but does not fully meet PLOS ONE’s publication criteria as it currently stands. Therefore, we invite you to submit a revised version of the manuscript that addresses the points raised during the review process.

Please carefully answer each of the critiquese provide by the reviewer in a point-by-point manner. 

We would appreciate receiving your revised manuscript by Feb 01 2020 11:59PM. To enhance the reproducibility of your results, we recommend that if applicable you deposit your laboratory protocols in protocols.io, where a protocol can be assigned its own identifier (DOI) such that it can be cited independently in the future. For instructions see: http://journals.plos.org/plosone/s/submission-guidelines#loc-laboratory-protocols

We look forward to receiving your revised manuscript.

Kind regards,

Xia Jin, MD, PhD

Academic Editor

PLOS ONE

Journal Requirements:

1.

"UMMC Medical Research Ethics Committee (MREC)

IRB reference number 926.4

Informed Consent obtained in writing and all patient data was anonymized"

Please amend your current ethics statement to confirm that your named institutional review board or ethics committee specifically approved this study.

3.  Please provide additional details regarding participant consent. In the ethics statement in the Methods and online submission information, please ensure that you have specified whether consent was written or verbal/oral. If consent was verbal/oral, please specify: 1) whether the ethics committee approved the verbal/oral consent procedure, 2) why written consent could not be obtained, and 3) how verbal/oral consent was recorded. If your study included minors, please state whether you obtained consent from parents or guardians in these cases.

Reviewers' comments:

Reviewer's Responses to Questions

**Comments to the Author**

1. Is the manuscript technically sound, and do the data support the conclusions?

Reviewer #1: Yes

2. Has the statistical analysis been performed appropriately and rigorously? 

Reviewer #1: I Don't Know

3. Have the authors made all data underlying the findings in their manuscript fully available?

Reviewer #1: No

4. Is the manuscript presented in an intelligible fashion and written in standard English?

Reviewer #1: Yes

5. Review Comments to the Author

Reviewer #1: This study by Soe and collegues presents a thorough characterization of the effect of patients’ sera with a broad range of disease severity, or NS1 antigen itself, on the integrity of microvascular endothelial cells. In the attempt of clarifying the ongoing debate on the role of NS1 in disrupting functional endothelial barrier, and its causal relation with Dengue disease progression, the authors further profiled the level of several cytokines and multiple metabolites using a multiplex flow cytometric assay and untargeted metabolomics, respectively. Overall the study is clearly written, results presented in an intelligible fashion and discussed in the context of the vast body of literature on the topic. Although no major breakthrough is presented, and most of the results are actually supportive of a lack of correlation between Dengue disease severity and any of the analysed variables (NS1 levels, pro-inflammatory cytokines or metabolites), the presented results are still important and worth publishing as they consolidate the notion that no specific/individual factor present in the sera (either of viral or cellular origin) can be clearly and unequivocally used as prognostic marker for disease severity. Furthermore, the study present a metabolic-based survey of patients sera, which have been well-characterized/normalized (i.e. based on NS1 levels and IgG), and therefore could be of relevance for the community.

Major points:

My only experimental concern relates to the ECIS experiment, with respect to the control used and the degree of details provided in the methods section.

More specifically (related to Fig.1-3):

- The purified NS1 protein used in Fig.1 was of which origin (i.e. bacterial, mammalian..etc), to which extent was purified, and in which diluent was diluted? This information is quite critical as it sets the ground for interpretation of the assay in the following figures. Furthermore, in order to exclude additional factors, the purified NS1 protein used here should have been diluted in healthy individual sera (perhaps this information is provided, though I could not find it in the text or the methods section).

- What are the positive ctrls used in this assay? (probably a known disruptor of endothelial barrier should be included here to show the sensitivity of the assay)

- What is the dynamic range of this assay? Without the ctrls described above, is difficult to assess whether the amplitude of the changes described in Fig. 1-3 is significant, and whether the variance across and within groups is acceptable.

Minor points:

Figures 7-10 could be easily grouped in one figure without losing clarity.

Furthermore it would be important for the reader, to include in the volcano plots and in the Venn diagrams (i.e. Fig.8-9) full names of at least some of the top metabolites identified as significantly regulated by one or more classes. These figures as such are currently not conveying any meaningful information.

Figures-7-13: The raw data and related statistical analysis of MS results should be presented in full in supplementary tables, to provide the reader with a mining-friendly resource. This is one of novelty points of the study, but it is virtually hidden as only selected candidates are presented in details in Fig.11-13.

6. PLOS authors have the option to publish the peer review history of their article (what does this mean?). If published, this will include your full peer review and any attached files.

Reviewer #1: Yes: Pietro Scaturro

---

## [Author Response · Author response to Decision Letter 0]

28 Jan 2020

Response to Reviewers:

Reviewers' comments:

Reviewer's Responses to Questions

Comments to the Author

1. Is the manuscript technically sound, and do the data support the conclusions?

Reviewer #1: Yes

Response: Thank you. 

2. Has the statistical analysis been performed appropriately and rigorously?

Reviewer #1: I Don't Know

Response: For cytokine profiling, the precision and reproducibility of the assays were assured by calculating the coefficient of variance of the replicates. Then we proceeded to access the significant differences of the cytokine expression profile between the categories using student’s T test. For untargeted metabolomics, all the statistical analysis was performed using the chemometrics software Mass Profiler Professional designed for the such purposes. 

3. Have the authors made all data underlying the findings in their manuscript fully available?

Reviewer #1: No

Response: The raw data of untargeted metabolomics have been prepared in full in supplementary S1 table (at end of manuscript).

 4. Is the manuscript presented in an intelligible fashion and written in standard English?

Reviewer #1: Yes

Response: Thank you. 

5. Review Comments to the Author

Reviewer #1: This study by Soe and collegues presents a thorough characterization of the effect of patients’ sera with a broad range of disease severity, or NS1 antigen itself, on the integrity of microvascular endothelial cells. In the attempt of clarifying the ongoing debate on the role of NS1 in disrupting functional endothelial barrier, and its causal relation with Dengue disease progression, the authors further profiled the level of several cytokines and multiple metabolites using a multiplex flow cytometric assay and untargeted metabolomics, respectively. Overall the study is clearly written, results presented in an intelligible fashion and discussed in the context of the vast body of literature on the topic. Although no major breakthrough is presented, and most of the results are actually supportive of a lack of correlation between Dengue disease severity and any of the analysed variables (NS1 levels, pro-inflammatory cytokines or metabolites), the presented results are still important and worth publishing as they consolidate the notion that no specific/individual factor present in the sera (either of viral or cellular origin) can be clearly and unequivocally used as prognostic marker for disease severity. Furthermore, the study present a metabolic-based survey of patients sera, which have been well-characterized/normalized (i.e. based on NS1 levels and IgG), and therefore could be of relevance for the community.

Major points:

My only experimental concern relates to the ECIS experiment, with respect to the control used and the degree of details provided in the methods section.

More specifically (related to Fig.1-3):

- The purified NS1 protein used in Fig.1 was of which origin (i.e. bacterial, mammalian..etc), to which extent was purified, and in which diluent was diluted? This information is quite critical as it sets the ground for interpretation of the assay in the following figures. 

Response: The purified NS1 protein is a recombinant dengue virus serotype 1 NS1 protein of sequence strain Nauru/Western Pacific/1974 expressed in 293 human cells. It is a purified recombinant protein with a C-terminal 6x His-tag and purity of >95% by SDS PAGE analysis diluted in buffer solution Dulbecco’s phosphate buffered saline, which was further diluted during the experiment using the respective media of each cell line used. The above information has been added to the methodology. 

Furthermore, in order to exclude additional factors, the purified NS1 protein used here should have been diluted in healthy individual sera (perhaps this information is provided, though I could not find it in the text or the methods section).

Response: Healthy individual served as negative control for the experiment and hence it contains no NS1 antigen. It was included to compare against the patients from the three categories of dengue. On the other hand, the purified NS1 protein was included not only to investigate the effect of viral factor (NS1) on endothelium, but also as a comparison for the different NS1 level present in the serum samples. The characteristics of all samples were included in table 1. 

- What are the positive ctrls used in this assay? (probably a known disruptor of endothelial barrier should be included here to show the sensitivity of the assay)

Response: A positive control was not included in the experiment as a standard disruptor of the endothelial barrier is yet to be established. We therefore included the purified NS1 protein to compare the modulation of endothelial barrier against patient sera that contain NS1 antigen at varying level with the presence of other host factors. 

- What is the dynamic range of this assay? Without the ctrls described above, is difficult to assess whether the amplitude of the changes described in Fig. 1-3 is significant, and whether the variance across and within groups is acceptable.

Response: The use of ECIS system allow us to monitor real-time cell barrier function, cell to cell adhesion and cell to matrix interactions, which is an incredibly sensitive (described below) method for evaluating barrier function of cells in vitro. It was used to compare the relative pattern of modulation between treated and untreated cells, rather than an absolute reading on the impedance changes. The results presented in this manuscript have been normalized and standardized to eliminate the effect of any gross influences during the experiment. The values were detected at multiple frequencies for three biological replicates with 10 detection points each, and the standard error of the mean was plotted as error bar in Fig. 1-3. The statistical measures included the normalization to a cell-free electrode, standardization of the fluctuations due to cell mobility such as micromotion, and a mathematical model that was implied to determine the changes in the endothelial barrier function. The readings were also normalized to the untreated endothelial cells and therefore, only values with significant difference as compared to the untreated cells were plotted in the graphs to represent the real-time changes of the barrier function of the treated cells. 

The following studies provide information on the sensitivity of ECIS system with regards to detecting low viral effects:

1. Mouse brain MEC infected with West Nile virus at MOI of 0.01 increased TEER up to 6 hours [1]

2. Reduced TEER was observed in Mouse Hepatitis Virus type 3-infected brain MEC at a MOI of 0.1 up to 72 hours [2]

3. Human Bocavirus-1 at the MOI of 0.001, 0.01, 0.1, 1, 10 and 100 reduced the TEER of human airway epithelial cells [3]

The studies above on the infection kinetics of different type of viruses on endothelial / epithelial cells provide information that the ECIS system is sensitive enough to detect small modulation of epithelial/endothelial cells infected with viruses at a relatively low MOIs. 

Limitation of the ECIS has also been included in the last paragraph of the discussion.

References quoted:

[1] Lazear et al. (2015) Interferon-λ restricts West Nile virus neuroinvasion by tightening the blood-brain barrier, Sci Transl Med, 7(284): 284ra59.

[2] Bleau et al. (2015) Brain Invasion by Mouse Hepatitis Virus Depends on Impairment of Tight Junctions and Beta Interferon Production in Brain Microvascular Endothelial Cells, J Virol, 89(19): 9896-9908. 

[3] Deng et al (2013) In Vitro Modeling of Human Bocavirus 1 Infection of Polarized Primary Human Airway Epithelia, J Virol, 87(7): 4097– 4102. 

Minor points:

Figures 7-10 could be easily grouped in one figure without losing clarity.

Response: Figures 7-9 have been combined into Figure 7 and be divided into parts i), ii) and iii). To better discuss the results for pathway analysis as a separated outcome, Figure 10 was kept as a single figure and has been changed to Figure 8. 

Furthermore it would be important for the reader, to include in the volcano plots and in the Venn diagrams (i.e. Fig.8-9) full names of at least some of the top metabolites identified as significantly regulated by one or more classes. These figures as such are currently not conveying any meaningful information.

Response: We agree with the reviewer’s comment, as such, the top regulated metabolites of each category have been added into the figures. 

Figures-7-13: The raw data and related statistical analysis of MS results should be presented in full in supplementary tables, to provide the reader with a mining-friendly resource. This is one of novelty points of the study, but it is virtually hidden as only selected candidates are presented in details in Fig.11-13.

Responses: The raw data of metabolites present in serum samples from each category of DWOWS, DWWS and SD has been presented in supplementary S1 table (at end of manuscript).

---

## [Decision Letter · Decision Letter 1]

5 Jun 2020

PONE-D-19-24309R1

Correlation of viral NS1 protein, host inflammatory cytokines and immune-related metabolites with disease severity of dengue virus infection

PLOS ONE

Dear Dr. Sekaran,

Thank you for submitting your manuscript to PLOS ONE. After careful consideration, we feel that it has merit but does not fully meet PLOS ONE’s publication criteria as it currently stands. Therefore, we invite you to submit a revised version of the manuscript that addresses the points raised during the review process.

Specifically try to answer the two reviewers comments about the manuscript organisation and the number of sample tested. As noted you results are in accordance with other studies that NS1 is not a potential marker for early diagnosis of dengue. 

your title might be: Correlation of host inflammatory cytokines and immune-related

metabolites, but not viral NS1 protein, with disease severity of dengue virus infection

We look forward to receiving your revised manuscript.

Kind regards,

Pierre Roques, Ph.D.

Academic Editor

PLOS ONE

Reviewers' comments:

Reviewer's Responses to Questions

**Comments to the Author**

1. If the authors have adequately addressed your comments raised in a previous round of review and you feel that this manuscript is now acceptable for publication, you may indicate that here to bypass the “Comments to the Author” section, enter your conflict of interest statement in the “Confidential to Editor” section, and submit your "Accept" recommendation.

Reviewer #2: (No Response)

Reviewer #3: All comments have been addressed

2. Is the manuscript technically sound, and do the data support the conclusions?

Reviewer #2: Yes

Reviewer #3: No

3. Has the statistical analysis been performed appropriately and rigorously? 

Reviewer #2: Yes

Reviewer #3: Yes

4. Have the authors made all data underlying the findings in their manuscript fully available?

Reviewer #2: No

Reviewer #3: No

5. Is the manuscript presented in an intelligible fashion and written in standard English?

Reviewer #2: Yes

Reviewer #3: Yes

6. Review Comments to the Author

Reviewer #2: In manuscript titled “Correlation of viral NS1 protein, host inflammatory cytokines and immune-related metabolites with disease severity of dengue virus infection”, the authors tested sera from 40 dengue patients, and intended to evaluate the relation between the presence of NS1/host cytokines/host metabolites and the severity of dengue diseases. The overall conclusion is that no significant correlation has been found between the two. Therefore their original intention to develop a potential marker for early diagnosis of dengue failed. The data collected in this manuscript can be valuable to understanding the dengue diseases and to the broader virology field if some of the essential controls are included in the experiments.

1) Introduction lacks a clear elaboration of the purpose of the study. It did not establish a clear rationale that supports the logic of testing NS1. For example, in lines 48-50, the authors said “Hence, this study aims to investigate the role of the viral non-structural 1 (NS1) protein, the inflammatory cytokines and the immune-related metabolites in contributing to the degree of severity of dengue virus infection”. However, the sentence before this conclusion is broad statement “The pathogenesis of severe dengue is theorized to be due to the intricate interactions between viral factors, host genetics and host immune activation” with no facts or details to explain why NS1 needs to be tested. More information should be included to elaborate the significance of why these experiments are done.

2) Some information introduced in the discussion, such as the first paragraph of discussion explaining ECIS, should be moved to introduction. It is strange to explain a method in discussion after the results have already been presented.

3) Figure 1 showed purified NS1 increases vascular permeability in pulmonary MEC. Figures 2-4 did not show substantial differences when they used different groups of patient sera on different cells. To examine whether NS1 has the potential to become a diagnostic marker for dengue, an experiment that dilute the purified NS1 in uninfected human sera would give a clear answer. It is quite possible that certain sera composition blocks the NS1 effect.

4) The authors did metabolites analysis with the patient sera. However, without a control of healthy human sera, these data have no meaning for future analysis. Another possible control is to use human sera from another viral infection to show whether conclusion drawn from Figure 7-10 is dengue specific or not.

Reviewer #3: The manuscript Correlation of viral NS1 protein, host inflammatory cytokines and immune-related metabolites with disease severity of dengue virus infection presented by Sekaran SD et al, is an interesting study regarding an important cause of disease, dengue.

It is a laboratory study that analyze the effect of the addition of Dengue sera to Electrical cell-substrate impedance senting (ECIS). Also, the citokynes and metabolic profile were measured.

I have several question.

1. The primaty and secondary dengue have differences in several geographic areas. The authors did no give any commentary about that.

2. The ECIS was performed with only 5 serum of dengue cases and none the original NS1 Ag was tested.

3. The cytokine profile were determine in only 40 serum (including from patients without alarm signs, with alarm signs or severe dengue). The authors need to include more patients.

4. The same observations are done regarding the metabolic expression in dengue patients.

7. PLOS authors have the option to publish the peer review history of their article (what does this mean?). If published, this will include your full peer review and any attached files.

Reviewer #2: No

Reviewer #3: Yes: Antonio Arbo, MD, MSc

---

## [Author Response · Author response to Decision Letter 1]

16 Jun 2020

Response to Reviewers:

Reviewer #2: In manuscript titled “Correlation of viral NS1 protein, host inflammatory cytokines and immune-related metabolites with disease severity of dengue virus infection”, the authors tested sera from 40 dengue patients, and intended to evaluate the relation between the presence of NS1/host cytokines/host metabolites and the severity of dengue diseases. The overall conclusion is that no significant correlation has been found between the two. Therefore, their original intention to develop a potential marker for early diagnosis of dengue failed. The data collected in this manuscript can be valuable to understanding the dengue diseases and to the broader virology field if some of the essential controls are included in the experiments.

1) Introduction lacks a clear elaboration of the purpose of the study. It did not establish a clear rationale that supports the logic of testing NS1. For example, in lines 48-50, the authors said “Hence, this study aims to investigate the role of the viral non-structural 1 (NS1) protein, the inflammatory cytokines and the immune-related metabolites in contributing to the degree of severity of dengue virus infection”. However, the sentence before this conclusion is broad statement “The pathogenesis of severe dengue is theorized to be due to the intricate interactions between viral factors, host genetics and host immune activation” with no facts or details to explain why NS1 needs to be tested. More information should be included to elaborate the significance of why these experiments are done.

Response: The role of NS1 in inducing vascular leakage have been described in the second paragraph. Nevertheless, the interactions between viral factors and host factors have been elaborated in the first paragraph to justify the rationale of investigating the correlation of viral factors and host factors with vascular leakage observed in patients with severe dengue. 

2) Some information introduced in the discussion, such as the first paragraph of discussion explaining ECIS, should be moved to introduction. It is strange to explain a method in discussion after the results have already been presented.

Response: The paragraph discussing the principle of ECIS has been moved to introduction. 

3) Figure 1 showed purified NS1 increases vascular permeability in pulmonary MEC. Figures 2-4 did not show substantial differences when they used different groups of patient sera on different cells. To examine whether NS1 has the potential to become a diagnostic marker for dengue, an experiment that dilute the purified NS1 in uninfected human sera would give a clear answer. It is quite possible that certain sera composition blocks the NS1 effect.

Response: The main objective of this manuscript is to assess the role of viral factor (NS1) and host factors (cytokines and metabolites) in contributing to the degree of vascular leakage in the microvascular endothelium during DENV infection, therefore, we focus on the sera of DENV infected patients with varying disease severity, which also contain NS1 protein at varying level. Since the findings showed that the serum NS1 level in dengue patients did not directly correlate to the extent of vascular leakage observed in MECs, we then proceeded to investigate the composition of the sera, in term of cytokines and metabolites, that might had overshadowed the effect of NS1. We have successfully identified some of the host proteins that were highly expressed in dengue patients and they might have played a bigger role as compared to NS1 in inducing vascular leakage. However, we would also like to highlight that no single factor alone should be concluded to be responsible for the progression of severe dengue, as the pathogenesis of dengue involves complex interactions between multiple factors. Nevertheless, the last paragraph of the introduction has been rephrased to better describe the main objectives of this study. 

4) The authors did metabolites analysis with the patient sera. However, without a control of healthy human sera, these data have no meaning for future analysis. Another possible control is to use human sera from another viral infection to show whether conclusion drawn from Figure 7-10 is dengue specific or not.

Response: Control of healthy human sera was included for all the experiments of ECIS, cytokines and metabolomics profiling. The expression of cytokines and metabolites in dengue patients was compared against the expression profile of healthy individuals. The methodology of sample collection and selection, as well as the results of samples characteristics have been revised to better explain the samples used in this study. 

 

Reviewer #3: The manuscript Correlation of viral NS1 protein, host inflammatory cytokines and immune-related metabolites with disease severity of dengue virus infection presented by Sekaran SD et al, is an interesting study regarding an important cause of disease, dengue.

It is a laboratory study that analyze the effect of the addition of Dengue sera to Electrical cell-substrate impedance sensing (ECIS). Also, the cytokines and metabolic profile were measured.

I have several questions.

1. The primary and secondary dengue have differences in several geographic areas. The authors did not give any commentary about that.

Response: The introduction has been revised to include the differences of primary and secondary infections in different geographic areas. 

2. The ECIS was performed with only 5 serum of dengue cases and none the original NS1 Ag was tested.

Response: For ECIS, we investigated the effect of direct stimulation of NS1 antigen on microvascular endothelium, as well as the effect of NS1 in the presence of other host factors in the patient sera towards microvascular endothelial barrier function. A total of 20 samples were used, 5 from each of the categories of healthy individual, dengue without warning signs, dengue with warning signs and severe dengue to provide an insight to the pattern of the changes in the barrier function of four microvascular endothelial cell lines in response to NS1/sera stimulation. 

3. The cytokine profile were determined in only 40 serum (including from patients without alarm signs, with alarm signs or severe dengue). The authors need to include more patients. The same observations are done regarding the metabolic expression in dengue patients.

Response: One of the limitations for this study was the small sample size of only 40 sera due to the difficulties in obtaining more samples from patients with severe dengue. Although the study has less statistical power, we were able to identify a few cytokines and metabolites that would provide insight on the role of host factors in the pathogenesis of dengue virus infection and possibly determine or predict disease severity that should be assessed with other approaches using a larger sample size in the future. This limitation has been discussed in the discussion.

---

## [Decision Letter · Decision Letter 2]

7 Jul 2020

PONE-D-19-24309R2

Correlation of host inflammatory cytokines and immune-related metabolites, but not viral NS1 protein, with disease severity of dengue virus infection

PLOS ONE

Dear Dr. Sekaran,

Thank you for submitting your manuscript to PLOS ONE. After careful consideration, we feel that it has merit but does not fully meet PLOS ONE’s publication criteria as it currently stands. Therefore, we invite you to submit a revised version of the manuscript that addresses the points raised during the review process.

The fact that at the end the studied population is very small taking in account all the variables as noted by the reviewers have to be indicated in the abstract as a limitation of the study. Some results confirmed previous studies from independant groups but new exploration done here remained exploratory and deserved to be confirmed or extended using independant cohort. All this limitation deserved to be highligted in the discussion but also in the conclusion of the abstract.

We look forward to receiving your revised manuscript.

Kind regards,

Pierre Roques, Ph.D.

Academic Editor

PLOS ONE

Additional Editor Comments (if provided):

Because the very low number of patients studied at the end the abstract should include this information. Ie the work is interesting but exploratory and deserved to be confirmed on independant cohort concerning the ECIS assay. This is mandatory

Reviewers' comments:

Reviewer's Responses to Questions

**Comments to the Author**

1. If the authors have adequately addressed your comments raised in a previous round of review and you feel that this manuscript is now acceptable for publication, you may indicate that here to bypass the “Comments to the Author” section, enter your conflict of interest statement in the “Confidential to Editor” section, and submit your "Accept" recommendation.

Reviewer #2: All comments have been addressed

Reviewer #3: All comments have been addressed

2. Is the manuscript technically sound, and do the data support the conclusions?

Reviewer #2: Yes

Reviewer #3: Partly

3. Has the statistical analysis been performed appropriately and rigorously? 

Reviewer #2: Yes

Reviewer #3: Yes

4. Have the authors made all data underlying the findings in their manuscript fully available?

Reviewer #2: Yes

Reviewer #3: Yes

5. Is the manuscript presented in an intelligible fashion and written in standard English?

Reviewer #2: Yes

Reviewer #3: Yes

6. Review Comments to the Author

Reviewer #2: 1) The authors addressed my questions. However, the newly added Figure 12 cannot be found in the paper.

2) Now that it is clearly described in the materials and methods that the 40 samples include 30 from patients and 10 from healthy people, the sample size is really small.

Reviewer #3: The manuscript Correlation of viral NS1 protein, host inflammatory cytokines and immune-related metabolites with disease severity of dengue virus infection presented by Sekaran SD et al, is an interesting study regarding an important

cause of disease, dengue.

It is a laboratory study that analyze the effect of the addition of Dengue sera to Electrical cell-substrate impedance sensing (ECIS). Also, the cytokines and metabolic profile were measured.

I have several questions.

Altough the authors say that they have include the differences of primary and secondary infections in different geographic areas, in the results the differences are scarse.

Moreover, the cytokibes were determined in only 40 serum samples from four categories, HC, DWOWS, DWWS and SD. This explained that only eight serum were analized by categorie.

The metabolic expresion of dengue patients have the same difficulty. Only 40 sera were studied, correspondind to foue categories.

7. PLOS authors have the option to publish the peer review history of their article (what does this mean?). If published, this will include your full peer review and any attached files.

Reviewer #2: No

Reviewer #3: **Yes: **Antonio Arbo, MD. Institute of Tropical Medicine. Asuncion, Paraguay

---

## [Author Response · Author response to Decision Letter 2]

18 Jul 2020

Response to Reviewers

Reviewer #2: 

1) The authors addressed my questions. However, the newly added Figure 12 cannot be found in the paper.

Response: Figure 12 was named after the graphical abstract; it is now being renamed as Graphical abstract and was uploaded separately. 

2) Now that it is clearly described in the materials and methods that the 40 samples include 30 from patients and 10 from healthy people, the sample size is really small.

Response: This limitation is now highlighted in the abstract to state that the findings deserved to be confirmed or extended using independent cohort with a bigger sample size. 

Reviewer #3: 

The manuscript Correlation of viral NS1 protein, host inflammatory cytokines and immune-related metabolites with disease severity of dengue virus infection presented by Sekaran SD et al, is an interesting study regarding an important cause of disease, dengue.

It is a laboratory study that analyzes the effect of the addition of Dengue sera to Electrical cell-substrate impedance sensing (ECIS). Also, the cytokines and metabolic profile were measured.

I have several questions.

1) Although the authors say that they have include the differences of primary and secondary infections in different geographic areas, in the results the differences are scarce.

Response: The differences of primary and secondary infections in different geographic areas were included in the introduction as supplementary information to the topic. However, we did not include the differences between primary and secondary infections in the experiment planning as we would like to focus on the categories of disease severity. 

2) Moreover, the cytokines were determined in only 40 serum samples from four categories, HC, DWOWS, DWWS and SD. This explained that only eight serum were analyzed by categories. The metabolic expressions of dengue patients have the same difficulty. Only 40 sera were studied, corresponding to four categories.

Response: The limitation of small sample size has been included in the discussion, as we feel that the exploratory findings merits further investigation, this limitation is now highlighted in the abstract to state that the findings deserved to be confirmed or extended using independent cohort with a bigger sample size.

---

## [Editor Report · Decision Letter 3]

22 Jul 2020

Correlation of host inflammatory cytokines and immune-related metabolites, but not viral NS1 protein, with disease severity of dengue virus infection

PONE-D-19-24309R3

Dear Dr. Sekaran,

We’re pleased to inform you that your manuscript has been judged scientifically suitable for publication and will be formally accepted for publication once it meets all outstanding technical requirements.

Kind regards,

Pierre Roques, Ph.D.

Academic Editor

PLOS ONE